# Clinical Benefit from Docetaxel +/− Ramucirumab Is Not Associated with Mutation Status in Metastatic Non-Small-Cell Lung Cancer Patients Who Progressed on Platinum Doublets and Immunotherapy

**DOI:** 10.3390/cancers16050935

**Published:** 2024-02-26

**Authors:** Kang Qin, Kaiwen Wang, Shenduo Li, Lingzhi Hong, Priyadharshini Padmakumar, Rinsurongkawong Waree, Shawna M. Hubert, Xiuning Le, Natalie Vokes, Kunal Rai, Ara Vaporciyan, Don L. Gibbons, John V. Heymach, J. Jack Lee, Scott E. Woodman, Caroline Chung, David A. Jaffray, Mehmet Altan, Yanyan Lou, Jianjun Zhang

**Affiliations:** 1Department of Thoracic/Head and Neck Medical Oncology, University of Texas MD Anderson Cancer Center, Houston, TX 77030, USA; kqin@mdanderson.org (K.Q.); lhong@mdanderson.org (L.H.); wcarter@mdanderson.org (R.W.); smhubert@mdanderson.org (S.M.H.); xle1@mdanderson.org (X.L.); nvokes@mdanderson.org (N.V.); dlgibbon@mdanderson.org (D.L.G.); jheymach@mdanderson.org (J.V.H.); maltan@mdanderson.org (M.A.); 2Division of Pharmacy, University of Texas MD Anderson Cancer Center, Houston, TX 77030, USA; kwang6@mdanderson.org; 3Division of Hematology and Oncology, Mayo Clinic, Jacksonville, FL 32224, USA; li.shenduo@mayo.edu; 4Department of Imaging Physics, University of Texas MD Anderson Cancer Center, Houston, TX 77030, USA; dajaffray@mdanderson.org; 5Department of Enterprise Data Engineering and Analytics, University of Texas MD Anderson Cancer Center, Houston, TX 77030, USA; ppadmakumar@mdanderson.org; 6Department of Genomic Medicine, University of Texas MD Anderson Cancer Center, Houston, TX 77030, USA; krai@mdanderson.org; 7Department of Thoracic and Cardiovascular Surgery, University of Texas MD Anderson Cancer Center, Houston, TX 77030, USA; avaporci@mdanderson.org; 8Department of Biostatistics, University of Texas MD Anderson Cancer Center, Houston, TX 77030, USA; jjlee@mdanderson.org; 9Department of Melanoma Medical Oncology, University of Texas MD Anderson Cancer Center, Houston, TX 77030, USA; swoodman@mdanderson.org; 10Department of Radiation Oncology and Diagnostic Imaging, University of Texas MD Anderson Cancer Center, Houston, TX 77030, USA; cchung3@mdanderson.org; 11Institute for Data Science in Oncology, University of Texas MD Anderson Cancer Center, Houston, TX 77030, USA

**Keywords:** metastatic non-small-cell lung cancer, docetaxel with or without ramucirumab, platinum–taxane

## Abstract

**Simple Summary:**

Docetaxel +/− ramucirumab is now frequently used as the standard chemotherapeutic regimen for patients with metastatic non-small-cell lung cancer (NSCLC) after progression on platinum doublets and immune checkpoint inhibitors (ICIs), regardless of the tumor histology. However, these regimens only lead to short-lived disease control with substantial toxicity, and there is an unmet need for more treatment options in this setting. Our study investigated whether the cancer gene mutation status is associated with clinical benefits from docetaxel +/− ramucirumab by analyzing treatment and outcomes by genomic status. We also explored whether platinum/taxane-based regimens offered a better clinical benefit in this patient population. The results of this study showed that the benefit from docetaxel +/− ramucirumab was not dependent on the cancer gene mutation status. Our exploratory analysis also suggested that platinum-/taxane-based regimens could be reasonable alternative treatment options with better efficacy and comparable tolerability.

**Abstract:**

Docetaxel +/− ramucirumab remains the standard-of-care therapy for patients with metastatic non-small-cell lung cancer (NSCLC) after progression on platinum doublets and immune checkpoint inhibitors (ICIs). The aim of our study was to investigate whether the cancer gene mutation status was associated with clinical benefits from docetaxel +/− ramucirumab. We also investigated whether platinum/taxane-based regimens offered a better clinical benefit in this patient population. A total of 454 patients were analyzed (docetaxel +/− ramucirumab n=381; platinum/taxane-based regimens n=73). Progression-free survival (PFS) and overall survival (OS) were compared among different subpopulations with different cancer gene mutations and between patients who received docetaxel +/− ramucirumab versus platinum/taxane-based regimens. Among patients who received docetaxel +/− ramucirumab, the top mutated cancer genes included TP53 (n=167), KRAS (n=127), EGFR (n=65), STK11 (n=32), ERBB2 (HER2) (n=26), etc. None of these cancer gene mutations or PD-L1 expression was associated with PFS or OS. Platinum/taxane-based regimens were associated with a significantly longer mQS (13.00 m, 95% Cl: 11.20–14.80 m versus 8.40 m, 95% Cl: 7.12–9.68 m, LogRank P=0.019) than docetaxel +/− ramcirumab. Key prognostic factors including age, histology, and performance status were not different between these two groups. In conclusion, in patients with metastatic NSCLC who have progressed on platinum doublets and ICIs, the clinical benefit from docetaxel +/− ramucirumab is not associated with the cancer gene mutation status. Platinum/taxane-based regimens may offer a superior clinical benefit over docetaxel +/− ramucirumab in this patient population.

## 1. Introduction

Lung cancer is the leading cause of cancer-related death worldwide [1]. Platinum-based chemotherapies have been the standard first-line treatments for patients with metastatic/ advanced-stage non-small-cell lung cancer (NSCLC) [2]. Immune checkpoint inhibitors (ICIs) targeting programmed cell death protein 1 (PD-1)/programmed death ligand 1 (PD-L1) produce an unrivaled durable clinical response, and first-line strategies for advanced NSCLC patients without a molecular driver have been shifted from traditional doublet chemotherapy to immunotherapy-based treatments with and without chemotherapy [3,4,5,6,7,8,9]. However, a durable response to ICIs is only achieved in a small subset of NSCLC patients, and most patients will develop resistance and disease progression [6,10,11,12].

For patients with metastatic NSCLC, who have progressed after platinum doublets and ICIs, subsequent therapy options include docetaxel (± ramucirumab), albumin-bound paclitaxel, gemcitabine, or pemetrexed (for nonsquamous only), depending on which agent has not been previously administered [13], among which docetaxel +/ramucirumab is the recommended salvage therapy regardless of tumor histology, based on the data of the REVEL study [14]. However, docetaxel +/− ramucirumab only leads to short-lived disease control and is associated with substantial toxicity. In the landmark REVEL study, the progression-free survival (PFS) was only 3.0 months for docetaxel monotherapy and 4.5 months for docetaxel + ramucirumab, while over 70% of patients had grade 3 or higher adverse events (79% for docetaxel + ramucirumab and 71% for docetaxel monotherapy) [14]. Therefore, precision patient selection and alternative salvage regimens are needed for this patient population.

Platinum-plus-taxane-based regimens have been well established as treatment options for metastatic NSCLC. Recently, multiple studies have investigated the clinical safety and efficacy of platinum/taxane in combination with ICIs [8,15,16,17,18]. The IMPOWER150 study demonstrated that the addition of atezolizumab to carboplatin/paclitaxel/bevacizumab chemotherapy improved the PFS and OS as the first- line treatment in patients with metastatic non-squamous NSCLC, without showing a detrimental effect on quality of life [18]. However, platinum/taxane-based regimens have not been systemically tested in the salvage setting in patients with metastatic NSCLC who have progressed on platinum doublets and ICIs. Another not fully addressed question is whether the cancer gene mutation status, which is known to profoundly impact the cancer biology and clinical presentation of NSCLCs, including the benefit from ICIs [19,20], impacts the benefits of docetaxel +/− ramucirumab. In this study, we investigated whether platinum/taxane-based regimens offered another salvage option in this patient population. we also sought to investigate whether different cancer gene mutation statuses were associated with different benefits of docetaxel +/ramucirumab in patients with metastatic NSCLC after progression on concomitant or sequential platinum-based chemotherapy and ICIs.

## 2. Materials and Methods

### 2.1. Study Population and Data Collection

The University of Texas MD Anderson Lung Cancer Moon Shot GEMINI (Genomic Marker-Guided Therapy Initiative) database contains information on consenting patients with lung cancer, including demographics, smoking history, treatment history, clinical outcomes, and tumor molecular profiling, etc. We queried the GEMINI database to identify patients who were treated at the MD Anderson Cancer Center between January 2009 and July 2023 who met the following criteria. (1) They had to be male/female patients (at least 18 years of age) with a histologically confirmed diagnosis of NSCLC including the following subtypes: adenocarcinoma, squamous cell carcinoma, adenosquamous carcinoma, non-small-cell carcinoma NOS (not otherwise specified), sarcomatoid carcinoma, large cell neuroendocrine carcinoma, etc. (2) Patients must have progressed on platinum-based chemotherapy (cisplatin 75 mg/$m^{2}$, day 1 + pemetrexed 500 mg/$m^{2}$, day 1, every 3 weeks; cisplatin 75 mg/$m^{2}$, day 1 + gemcitabine 1000 mg/$m^{2}$ or 1250 mg/$m^{2}$, days 1 and 8, every 3 weeks; cisplatin 75 mg/$m^{2}$, day 1 + docetaxel 75 mg/$m^{2}$, day 1, every 3 weeks; cisplatin 50 mg/$m^{2}$, days 1 and 8 + vinorelbine 25 mg/$m^{2}$, days 1, 8, 15, and 22, every 4 weeks; cisplatin 100 mg/$m^{2}$ day 1 + vinorelbine 30 mg/$m^{2}$ days 1, 8, 15, and 22, every 4 weeks; cisplatin 75–80 mg/$m^{2}$, day 1 + vinorelbine 25–30 mg/$m^{2}$, days 1 and 8, every 3 weeks; cisplatin 100 mg/$m^{2}$ day 1 + etoposide 100 mg/$m^{2}$, days 1–3, every 4 weeks; carboplatin AUC 5 or AUC 6, day 1 + paclitaxel 175 mg/$m^{2}$ or 200 mg/$m^{2}$, day 1, every 3 weeks; carboplatin AUC 5 or AUC 6, day 1 + gemcitabine 1000 mg/$m^{2}$, days 1 and 8, every 3 weeks; carboplatin AUC 5 or AUC 6, day 1 + pemetrexed 500 mg/$m^{2}$, day 1 every 3 weeks, etc.) and any of the FDA-approved PD-1 or PD-L1 immune checkpoint inhibitors, either given sequentially or in combination. (3) Patients had to have received docetaxel monotherapy (75 mg/$m^{2}$, day 1, every 3 weeks), docetaxel plus ramucirumab (docetaxel 75 mg/$m^{2}$, day 1 + ramucirumab 10 mg/kg, day 1, every 3 weeks), or platinum/taxane-based regimens (a. carboplatin AUC 5 or AUC 6, day 1 + paclitaxel 175 mg/$m^{2}$ or 200 mg/$m^{2}$, day 1 + atezolizumab 1200 mg, day 1 + bevacizumab 15 mg/kg, day 1, every 3 weeks; b. carboplatin AUC 5 or AUC 6, day 1 + paclitaxel 175 mg/$m^{2}$ or 200 mg/$m^{2}$, day 1 + bevacizumab 15 mg/kg, day 1, every 3 weeks; c. carboplatin AUC 5 or AUC 6, day 1 + paclitaxel 175 mg/$m^{2}$ or 200 mg/$m^{2}$, day 1 + atezolizumab 1200 mg, day 1, every 3 weeks; d. carboplatin AUC 5 or AUC 6, day 1 + docetaxel 75 mg/$m^{2}$, day 1, every 3 weeks + pembrolizumab 200 mg, day 1, every 3 weeks or 400 mg, every 6 weeks; e. carboplatin AUC 5 or AUC 6, day 1 + paclitaxel 150 mg/$m^{2}$, day 1, every 3 weeks + pembrolizumab 200 mg, day 1, every 3 weeks or 400 mg, day 1, every 6 weeks; f. carboplatin AUC 5 or AUC 6, day 1 + paclitaxel 150 mg/$m^{2}$, day 1 + nivolumab 360 mg, day 1, every 3 weeks; g. cisplatin 75–100 mg/$m^{2}$, day 1 + paclitaxel 150 mg/$m^{2}$, day 1, every 3 weeks + pembrolizumab 200 mg, day 1, every 3 weeks or 400 mg, day 1, every 6 weeks; h. carboplatin AUC 5 or AUC 6, day 1 + docetaxel 75 mg/$m^{2}$, day 1, every 3 weeks; i. carboplatin AUC 5 or AUC 6, day 1 + paclitaxel 150 mg/$m^{2}$, day 1, every 3 weeks) after progression on concomitant or sequential platinum doublets and ICIs. (4) For patients with squamous carcinoma, who received platinum/taxane and immunotherapy sequentially, they must have had platinum doublets as a first-line therapy followed by immunotherapy. For patients with squamous carcinoma, who received concurrent platinum/taxane and immunotherapy, they must have progressed on maintenance immunotherapy. (5) Patients had to have baseline imaging and at least one repeated radiological examination. (6) Patients had to be at stage IV NSCLC when each treatment mentioned above started. The baseline characteristics of the patients, including age, gender, smoking history, family history, histology, Eastern Cooperative Oncology Group (ECOG) performance status when treatment started, baseline metastatic sites, cancer gene mutation status, PDL1 expression, prior treatment history, etc., were obtained for data analysis. PD-L1 and genomic profiling data were collected from pathology reports, as previously reported [19,21]. An external cohort that met the inclusion criteria from the Mayo Clinic was also included for analysis. This study was approved by the institutional review board at the MD Anderson Cancer Center and Mayo Clinic. Written informed consent was obtained from all patients at the MD Anderson Cancer Center and was waived due to the retrospective nature of the study for patients at the Mayo Clinic. The study was conducted in accordance with the Declaration of Helsinki.

### 2.2. Clinical Endpoints

Real-world PFS and OS were primary outcomes for this study. PFS was defined as the time of initiation of each treatment until disease progression or death (whichever occurred first). Disease progression was determined based on pathologic confirmation or clinical progression determined by the treating physician based on imaging reports or through clinical assessment. OS was measured from the date of initiation of each treatment to the date of death from any cause.

### 2.3. Statistical Analysis

Standard descriptive statistics such as the median and range, frequencies, and percentages were used for the baseline characteristics of the patients. Continuous variables were summarized using medians and interquartile ranges. Categorical variables were calculated as frequencies or percentages. The Kaplan–Meier method was used for the estimation of PFS and OS, and differences were compared through the log-rank test. Cox proportional hazards regression models were used to evaluate the associations between clinical–genomic factors and PFS/OS; hazard ratios (HRs) and 95% confidence intervals (Cls) were obtained. Clinical features were selected for multivariate analysis if they were significant in the univariate analysis for PFS and OS. Statistical analyses for continuous and categorical variables and comparisons of the characteristics between the two groups, as well as the associations between the mutational status and response, were assessed by Student’s *t*-test, the Mann–Whitney U test, Pearson’s and Spearman’s chi-squared tests, or the Fisher exact test, as appropriate. In the exploratory subgroup analysis of the overall population, Cox proportional hazard regression models were used to adjust for relevant clinicopathological variables in the multivariable analysis. Statistical analyses were performed using SPSS version 29.0 and GraphPad Prism version 9.0.

## 3. Results

### 3.1. Patient Characteristics

At the data-lock date of 14 July 2023, a total of 454 patients meeting the inclusion criteria were identified: 178 patients (39.21%) received docetaxel monotherapy, 203 patients (44.71%) received docetaxel + ramucirumab, and 73 patients (16.08%) received platinum/taxane-based regimens (Figure 1, Appendix A). The baseline demographic characteristics are summarized in Table 1. Across the entire cohort, the median age was 64 years (range 27–90 years), 231 (50.88%) patients were male, 330 (72.69%) had a history of smoking, 371 (81.72%) patients had adenocarcinoma, 188 (41.41%) patients were treated in the second-line setting after progression on concurrent chemotherapy/immunotherapy, and 266 (58.59%) patients were treated in the third-line or later setting. Patients with targetable mutations had exhausted the targeted therapy options. The majority of patients (*n* = 349, 76.87%) had an Eastern Cooperative Oncology Group (ECOG) performance status (PS) of 0 or 1 at the time of starting of each treatment. Across the entire cohort, the median treatment duration was 1.80 m (0.70 m–32.9 m). The median duration of treatment with docetaxel monotherapy, docetaxel + ramucirumab, and platinum/taxane-based regimens was 1.60 m (0.70 m–20.23 m), 1.90 m (0.70 m–32.90 m), and 3.19 m (0.70 m–33.47 m), respectively. By the data-lock date on 14 July 2023, 421 patients had experienced disease progression (7.27% censoring rate) and 130 patients had died (28.63% censoring rate).

Among the 381 patients who received docetaxel +/− ramucirumab, the mPFS was 3.80 m (95 % Cl: 3.31 m–4.29 m) and the mOS was 8.40 m (95 % Cl: 7.12 m–9.68 m). Likely due to the small sample size, we did not observe significant differences in mPFS (3.25 m versus 4.18 m, LogRank *p* = 0.13; HR = 0.85, 95 % CI: 0.69 m–1.05 m, *p* = 0.13) or mOS (7.80 m vs. 8.93 m, *p* = 0.21; HR = 0.86, 95 % CI: 0.68–1.09, *p* = 0.21) between patients who were treated with docetaxel monotherapy and docetaxel + ramucirumab (Appendix A), although the ramucirumab group had a numerically longer PFS and OS. The univariate analysis identified that the female gender and an ECOG of 0–1 were associated with significantly longer mPFS (4.01 m for females versus 3.50 m for males, HR = 0.81, 95 % CI: 0.65–1.00, *p* = 0.045; 4.01 m for ECOG 0–1 versus 3.20 m for ECOG 2–3, HR = 0.78, 95 % CI: 0.61–1.00, *p* = 0.05) and longer mOS (9.76 m for females versus 7.33 m for males, HR = 0.76, 95 % CI: 0.60–0.96, *p* = 0.022; 8.93 m for ECOG 0–1 versus 6.64 m for ECOG 2–3, HR = 0.59, 95 % CI: 0.45–0.78, *p* < 0.001). An adenocarcinoma histology was also associated with favorable mOS (9.76 m for females versus 7.33 m for males, HR 0.76, 95% CI: 0.60–0.96, *p* = 0.02) (Appendix A). The subsequent multivariate analysis confirmed that that the female gender and ECOG 0–1 were independent favorable prognostic factors of PFS and OS (Appendix A).

In the subgroup of patients treated with docetaxel monotherapy (*n* = 178), baseline adrenal metastasis was an independent negative prognostic factor of PFS (2.87 m vs. 3.68 m, HR = 0.64, 95 % CI 0.44–0.94, *p* = 0.02), and ECOG 0–1 (8.40 m vs. 6.64 m, *p* = 0.003; HR = 0.57, 95 % CI: 0.39–0.83, *p* = 0.004) was an independent favorable factor of OS in the univariate analysis. Moreover, in the subgroup of patients treated with docetaxel + ramucirumab (*n* = 203), the female gender (4.83 m vs. 3.83 m, HR = 0.78, 95 % CI 0.66–0.89, *p* = 0.006) and ECOG 0–1 (6.73 m vs. 9.30 m, HR = 0.63, 95 % CI: 0.42–0.94, *p* = 0.022) were independent favorable factors for OS (Appendix A).

### 3.2. Neither Genomic Subtype nor PD-L1 Expression Was Associated with Benefit from Docetarel+/Ramucirumab

Among all patients treated with docetaxel +/− ramucirumab (*n* = 381), the genomic status of 28 (7.35 %) patients was unknown. The top mutated cancer genes in this cohort were TP53 (*n* = 167), KRAS (*n* = 127), EGFR (*n* = 65), STK11 (*n* = 32), ERBB2 (HER2) (*n* = 26), MET (*n* = 24), PIK3CA alterations (*n* = 19), etc. (Table 2). None of the cancer gene mutations was associated with the clinical efficacy of docetaxel monotherapy, docetaxel + ramucirumab, or the overall cohort treated with docetaxel +/− ramucirumab in this patient population (Appendix A). We next focused on the adenocarcinomas and looked into the common cancer gene mutations in EGFR and KRAS. Neither EGFR nor KRAS mutations were associated with a benefit from docetaxel monotherapy or docetaxel + ramucirumab (Appendix A).

Among the patients treated with docetaxel +/− ramucirumab, PD-L1 expression was low (TPS < 1%), intermediate (TPS 1–49%), high (TPS ≥ 50%), and unknown in 132 (34.65%), 119 (31.23%), 68 (17.85%), and 62 (16.27%) patients, respectively (Appendix A). PD-L1 expression was not associated with either PFS or OS (Appendix A).

### 3.3. Platinum/Taxane-Based Regimens Had Superior Efficacy to Docetaxel +/− Ramucirumab

We next sought to compare docetaxel +/− ramucirumab with platinum/taxane-based regimens. A numerically longer PFS (mPFS: 5.16 m 95 % CI: 4.09 m–6.23 m in platinum/taxane group versus 3.80 m 95% CI: 3.31 m–4.29 m in docetaxel +/− ramucirumab group, LogRank *p* = 0.092) and a significantly longer OS (mOS: 13.00 m 95% CI: 11.20 m–14.80 m versus 8.40 m 95% CI: 7.12 m–9.68 m in docetaxel +/− ramucirumab group, LogRank *p* = 0.019) were observed in the platinum/taxane group compared to the docetaxel +/− ramcirumab group. Notably, the difference remained significant in patients who developed disease progression 2–4 months after discontinuing the initial platinum following platinum doublet and ICI induction (mOS: 13.90 m, 95% CI: 12.59 m–15.22 m in platinum/taxane group versus 6.70 m, 95% CI: 4.02 m–9.38 m docetaxel +/− ramcirumab group, LogRank *p* = 0.002) (Figure 2). Importantly, there were no significant differences in gender, ECOG status, baseline adrenal metastasis, or baseline brain metastasis between the platinum/taxane-based regimen group and docetaxel +/− ramcirumab group (Appendix A). A total of 59 patients discontinued treatment due to adverse events, including 11.2% (*n* = 20) of the patients who received docetaxel monotherapy, 13.8% (*n* = 28) of the patients who received docetaxel + ramucirumab, and 15.1% (*n* = 11) of the patients who received platinum/taxane, respectively. The proportion of patients who discontinued treatment due to adverse effects was not different among these groups (*p* = 0.64), suggesting that the tolerability was not inferior for platinum/taxane-based regimens in this patient population (Figure 2).

## 4. Discussion

The management of patients with metastatic NSCLC, who have progressed on platinum doublets and ICIs, remains an unmet clinical need. These ongoing efforts entail biomarker-based patient selection for existing SOC treatment options and the development of novel therapeutic strategies. Docetaxel +/− ramucirumab is still the most used chemotherapy regimen in this population and there are currently no reliable biomarkers to identify patients who may or may not benefit from these treatments. Cancer gene mutations and PD-L1 levels are known to be associated with the benefit from various therapies, which promoted us to investigate whether these biomarkers are associated with benefits from docetaxel +/− ramucirumab.

PD-L1 is the most used biomarker to guide decisions regarding ICI regimens for patients with metastatic NSCLC, and it has been reported to be positively associated with a benefit from ICIs [19,20]. Multiple studies have investigated the role of PD-L1 expression in the survival of patients who receive docetaxel monotherapy or docetaxel + ramucirumab. In a study by Yoshimura et al., PD-L1 expression was not found to be associated with the efficacy of docetaxel + ramucirumab in NSCLC patients who progressed on platinum-based chemotherapy [22]. In our study, the benefit from docetaxel +/− ramucirumab was not associated with the level of PD-L1 expression in patients who had progressed on platinum doublets and ICIs. Taken together, these results indicate that the PD-L1 expression status may not impact docetaxel +/− ramucirumab in the salvage setting and novel predictive biomarkers are needed in this patient population.

The discovery of actionable mutations has significantly advanced precision oncology for lung cancer patients [23]. The presence of these mutations is not only predictive of a superior benefit from small-molecule tyrosine kinase inhibitors (TKI) [24,25,26,27,28,29,30,31,32,33,34,35,36,37,38,39], but also correlates with a clinical benefit from chemotherapy [40,41,42,43] and ICIs [20,44,45,46]. The association of these mutations with a clinical benefit from docetaxel +/− ramucirumab remains to be elucidated. It is important to note that ramucirumab has been shown to improve the efficacy of EGFR TKI erlotinib in EGFR-mutant NSCLC in the first-line setting [47], in line with the hypothesis of an angiogenesis dependency in EGFR-mutant lung cancers. Recently, a study by Furuya et al. also revealed that the EGFR mutation status (*n* = 24 for EGFR mutant and *n* = 88 for EGFR wild type) might be a positive prognostic factor for PFS in NSCLC patients treated with docetaxel + ramucirumab (mPFS was 5.7 months for the EGFR mutant group compared with 3.6 months for the EGFR wild-type group, HR 0.53, 95 % Cl 0.32–0.87; *p* = 0.01) [48]. On the other hand, in the landmark REVEL study, a small number of patients with known EGFR mutations (*n* = 33) were included, and the impact of ramucirumab on PFS and OS was present for both the EGFR-mutated (*n* = 33, PFS, HR: 0.64, 95% Cl: 0.31–1.32; OS HR: 0.79, 95% Cl: 0.34–1.83) and EGFR wild-type patients (*n* = 404, PFS HR: 0.77, 95% Cl: 0.63–0.95; OS HR: 0.83, 95% Cl: 0.65–1.05) [14]. However, all these studies were limited by their small sample sizes. To the best of our knowledge, our study provides data on the largest NSCLC cohort with EGFR mutations (*n* = 65) who had progressed on TKIs, platinum doublets, and ICIs. No association was found between EGFR mutations and the benefit from docetaxel +/− ramucirumab, which was consistent with the findings of Ellis-Caleo et al. [49]. Additionally, none of the common genomic alterations in TP53, KRAS, EGFR, BRAF, STK11, ERBB2, etc., was associated with a benefit from docetaxel +/− ramucirumab.

In the search for alternative treatments to docetaxel +/− ramucirumab in NSCLC patients who had progressed on platinum doublets and ICIs, we found that platinum/taxane-based regimens provided a numerically longer PFS (PFS: 3.80 m 95% CI: 3.31 m–4.29 m versus 5.50 m 95% Cl: 4.19 m–6.81 m, *p* = 0.092) and significantly longer OS (mOS: 8.40 m 95% Cl: 7.12 m–9.68 m versus 13.00 m 95% Cl: 11.20 m–14.80 m *p* = 0.019), suggesting that platinum/taxane-based regimens could be one of the most promising strategies to further improve NSCLC patients’ outcomes after progressing on first-line chemoimmunotherapy.

Platinum agents exert their therapeutic effect by forming covalent bonds with DNA, leading to the creation of DNA cross-links that hinder DNA replication, ultimately resulting in cell cycle arrest and the cessation of tumor cell proliferation [50]. In contrast, taxanes such as paclitaxel and docetaxel disrupt the cell cycle by binding to the β tubulin subunit, stabilizing microtubules, inducing mitotic arrest, and ultimately triggering apoptosis in tumor cells [51,52,53]. The combination of platinum and taxane agents is widely used in clinical practice due to their additive and synergistic effects, as supported by both in vitro data and clinical studies. Phase III clinical trials have demonstrated that this combination therapy extends the median survival to 8–11 months and yields 1-year survival rates ranging from 31% to 46% [54,55,56,57]. Despite this, there is a lack of systematic studies evaluating platinum/taxane-based regimens in advanced NSCLC patients who have progressed on previous platinum doublets and immunotherapy. Our data indicate promising clinical efficacy in this patient population, suggesting that the additive and synergistic effects between platinum and taxane agents may still be relevant in this context.

Another plausible explanation is that platinum/taxane-based regimens may only be offered to patients who are healthier (better PS), have better prognostic factors, or have tumors that are very sensitive to platinum. However, our data demonstrated that there was no significant difference in key prognostic factors such as age, gender, ECOG PS, and baseline metastatic patterns between patients who received SOC docetaxel +/ramucirumab versus those treated with platinum/taxane-based regimens. Additionally, in the subgroup of patients who had disease progression 2–4 months after discontinuing platinum, platinum/taxane-based regimens still demonstrated significantly longer OS than docetaxel +/− ramucirumab, indicating that the observed superior benefit from platinum/taxane-based regimens was unlikely due to a greater portion of patients with platinum-sensitive tumors in this group. Importantly, the rate of treatment discontinuation due to adverse effects was also no different between these two regimen groups. Taken together, these data suggest that in patients with metastatic NSCLC who have progressed on platinum doublets and ICIs, platinum/taxane-based regimens could be reasonable options with comparable tolerability and possibly better clinical efficacy compared to SOC docetaxel +/− ramucirumab.

As a real-world study, our work has important limitations. First, due to its retrospective nature, we were limited by the inadequate control for potential confounding factors that may have impacted the clinical benefit, such as metastatic patterns and prior treatment history. Second, the sample size was still relatively small, which might be the reason for the lack of significance in PFS and OS between the docetaxel + ramucirumab group and the docetaxel monotherapy group. Third, due to the relatively small sample size, we included all eligible patients without a pre-determined power analysis to detect differences between subgroups. Therefore, this study serves as an exploratory analysis of overall outcomes, rather than a definitive assessment of the differences among specific subgroups. Fourth, all patients in this study were treated at comprehensive cancer centers, which may limit the generalizability of our findings. Real-world practices can vary significantly between academic centers and community settings. While academic centers often offer access to cutting-edge treatments and clinical trials, community practices may serve a more diverse patient population with varying levels of access to resources and specialized care. Therefore, the findings from our study may not fully reflect the diversity of real-world clinical practice. Moreover, differences in patient demographics, genetic factors, and healthcare infrastructure could also impact the generalizability of our study findings. It is essential to consider these factors when interpreting and applying our results to different patient populations. On the other hand, while randomized clinical trials are the gold standard in providing evidence in clinical practice, conducting such trials may not always be practical, especially in certain circumstances. For example, although our data demonstrated the superiority of platinum/taxane-based regimens over docetaxel +/− ramucirumab, conducting randomized trials to directly compare these two regimens in the salvage setting can be challenging both practically and financially, and biomarker-based novel agents may offer a more feasible approach to improve the clinical outcomes of this patient population. Real-world data can also provide valuable insights to guide treatment decision making until novel trials change the standard-of-care practice.

## 5. Conclusions

To our knowledge, this is the largest multi-institute real-world study that has systematically investigated the associations of different molecular alterations and the benefits obtained from the most prescribed SOC regimen, docetaxel +/− ramucirumab, in metastatic NSCLC patients after progression on concomitant or sequential platinum-based chemotherapy and ICIs. Furthermore, our analyses suggest that platinum/taxane-based regimens may be an option in this patient population, with comparable tolerability and possibly better clinical efficacy compared to docetaxel +/− ramucirumab.

## Figures and Tables

**Figure 1 cancers-16-00935-f001:**
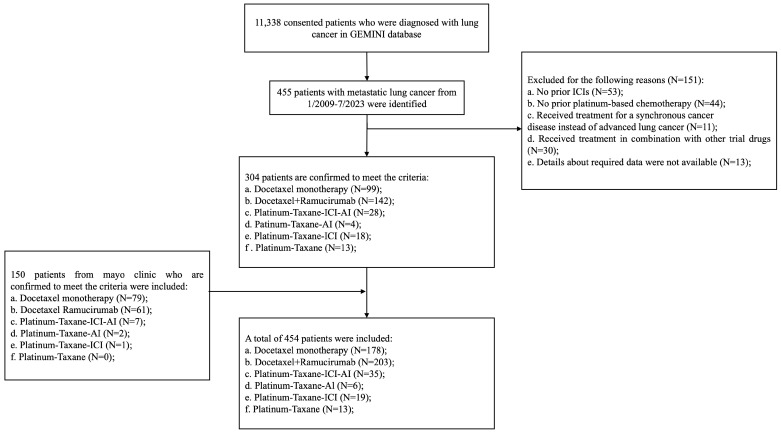
Consort diagram of study. Abbreviations: ICI, immune checkpoint inhibitors; AI, anti-angiogenesis inhibitor; GEMINI, Genomic Marker-Guided Therapy Initiative.

**Figure 2 cancers-16-00935-f002:**
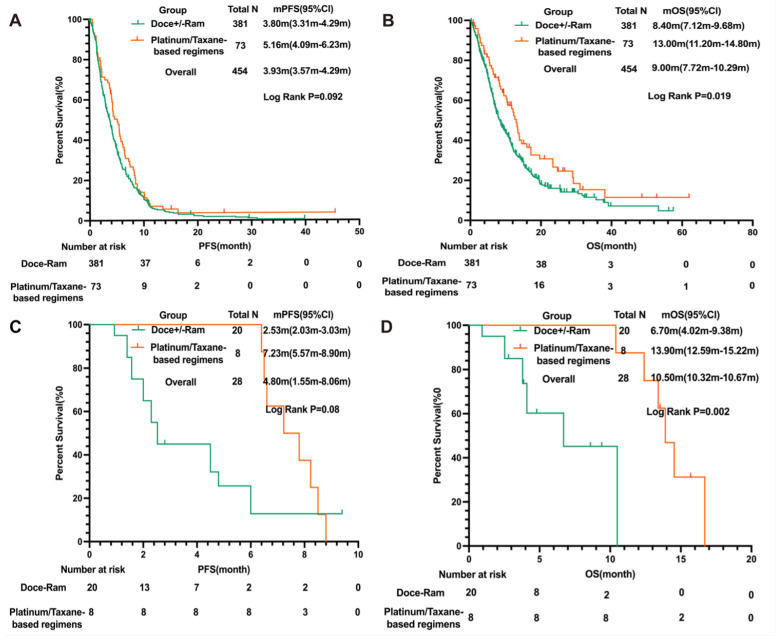
Comparison of PFS and OS in patients treated with docetaxel +/− ramucirumab versus platinum/taxane-based regimens. (**A**) PFS and (**B**) OS of patients treated with docetaxel + ramucirumab (N=381) versus platinum/taxane-based regimens (N=73); (**C**) PFS and (**D**) OS of patients treated with docetaxel + ramucirumab (N=20) versus platinum/taxane-based regimens (N=8) who developed disease progression within 2–4 months of initial platinum discontinuation following platinum doublet and ICI induction in the MDA cohort. Abbreviations: PFS, progression-free survival; OS, overall survival; HR, hazard ratio; CI, confidence interval; Doce, docetaxel; Ram, ramucirumab; ICI, immune checkpoint inhibitor.

**Table 1 cancers-16-00935-t001:** Baseline clinicopathological characteristics.

Clinicopathological Characteristics	Overall (*n* = 454)	Docetaxel (*n* = 178)	Docetaxel + Ramucirumab (*n* = 203)	Platinum-Taxane-AIs-ICIs (*n* = 35)	Platinum-Taxane-AIs (*n* = 6)	Platinum-Taxane-ICIs (*n* = 19)	Platinum-Taxane (*n* = 13)
**Age, years**							
Median	64	65	64	59	56.5	64	67
Range	27–90	31–90	27–82	28–74	48–71	46–76	46–76
**Gender**							
Male	231 (50.88)	88 (49.44)	107 (52.71)	14 (40)	2 (33.33)	14 (73.68)	6 (46.15)
Female	223 (49.12)	90 (50.56)	96 (47.29)	21 (60)	4 (66.67)	5 (26.32)	7 (53.85)
**Smoking History**							
Never	124 (27.31)	45 (25.28)	55 (27.09)	14 (40)	2 (33.33)	3 (15.80)	5 (38.46)
Current	38 (8.37)	16 (8.99)	15 (7.39)	2 (5.71)	1 (16.67)	4 (21.05)	0 (0)
Former	292 (64.32)	117 (65.73)	133 (65.52)	19 (54.29)	3 (50)	12 (63.16)	8 (61.54)
**ECOG PS**							
0–1	349 (76.87)	132 (74.16)	162 (79.80)	26 (74.29)	6 (100)	13 (68.42)	10 (76.92)
2–3	105 (23.13)	46 (25.84)	41 (21.20)	9 (25.71)	0 (0)	6 (31.58)	3 (23.08)
**Clinical Stage**							
IIIB	5 (1.10)	2 (1.12)	0 (0)	1 (2.86)	0 (0)	1 (5.26)	1 (7.69)
IVA	71 (15.64)	32 (17.98)	28 (13.79)	2 (5.71)	0 (0)	6 (31.58)	3 (23.08)
IVB	378 (83.26)	144 (80.90)	175 (86.21)	32 (91.43)	6 (100)	12 (63.16)	9 (69.23)
**Race**							
White or Caucasian	374 (82.38)	151 (84.83)	162 (79.80)	29 (82.86)	4 (60.0)	16 (84.21)	12 (92.31)
Asian	34 (7.27)	12 (6.74)	15 (0.49)	3 (8.57)	1 (20.0)	2 (10.53)	1 (7.69)
Black or African American	34 (7.49)	11 (6.18)	20 (9.85)	1 (2.86)	1 (20.0)	1 (5.26)	0 (0)
Hispanic or Latino	12 (2.64)	4 (2.25)	6 (2.96)	2 (5.71)	0 (0)	0 (0)	0 (0)
**Histology**							
ADC	371 (81.72)	145 (81.46)	171 (84.24)	34 (97.14)	5 (83.33)	6 (31.58)	10 (76.92)
SQC	64 (14.10)	26 (14.61)	24 (11.82)	0 (0)	0 (0)	11 (57.89)	3 (23.08)
Adenosquamous	5 (1.10)	3 (1.69)	2 (0.99)	0 (0)	0 (0)	0 (0)	0 (0)
Carcinoid tumor	1 (0.22)	1 (0.56)	0 (0)	0 (0)	0 (0)	0 (0)	0 (0)
Large cell neuroendocrine carcinoma	8 (1.76)	2 (1.12)	4 (2.25)	0 (0)	1 (16.67)	1 (5.26)	0 (0)
NSCLC—nonspecific subtype	5 (1.10)	1 (0.56)	2 (0.99)	1 (2.86)	0 (0)	1 (5.26)	0 (0)
**Recurrent NSCLC**							
Yes	63 (13.88)	32 (17.98)	26 (12.81)	3 (8.57)	1 (16.67)	1 (5.26)	0 (0)
No	391 (86.12)	146 (82.02)	177 (87.19)	32 (91.43)	5 (83.33)	18 (94.74)	13 (100.00)
**Line of Treatment**							
2	188 (41.41)	65 (36.52)	92 (45.32)	14 (40.00)	3 (50.00)	8 (42.11)	6 (46.15)
3	144 (31.72)	63 (35.39)	63 (31.03)	8 (22.86)	1 (16.67)	6 (31.58)	3 (23.08)
4	68 (14.98)	29 (16.29)	24 (11.82)	6 (17.14)	1 (16.67)	5 (26.32)	3 (23.08)
5–10	54 (11.89)	21 (11.80)	24 (11.82)	7 (20.00)	1 (16.67)	0 (0)	1 (7.69)
**Metastasis at Baseline**							
Brain	175 (39.95)	69 (38.76)	85 (41.87)	14 (40.00)	3 (50)	5 (26.32)	2 (15.38)
Liver	105 (23.97)	38 ( (21.35)	52 (25.62)	11 (31.43)	2 (33.33)	3 (15.79)	4 (30.77)
Bone	241 (55.02)	103 (57.87)	114 (56.16)	16 (45.71)	5 (83.33)	5 (26.32)	6 (46.15)
Adrenal	69 (15.75)	35 (19.66)	29 (14.29)	5 (14.29)	1 (16.67)	2 (10.53)	0 (0)
TX duration	1.80 (0.70–32.9)	1.60 (0.70–20.23)	1.90 (0.70–32.90)	5.30 (0.70–33.47)	3.50 (0.70–8.30)	3.40 (0.63–14.07)	1.60 (0.63–5.10)
**Reason for Treatment Discontinuation**							
Disease progression/death	338 (77.17)	143 (80.33)	156 (76.85)	27 (77.14)	5 (83.33)	14 (73.68)	4 (30.77)
Toxicity	59 (13.47)	20 (11.24)	28 (13.79)	4 (11.43)	1 (16.67)	1 (5.26)	5 (38.46)
Other treatment plans available	7 (1.60)	2 (1.12)	3 (1.48)	0 (0)	0 (0)	1 (5.26)	2 (15.38)
Ongoing treatment	5 (1.14)	2 (1.12)	1 (0.49)	1 (2.86)	0 (0)	2 (10.53)	1 (7.69)
Loss to follow-up	9 (2.05)	2 (1.12)	6 (2.96)	1 (2.86)	0 (0)	0 (0)	0 (0)
TX completed	20 (4.57)	9 (5.06)	9 (4.43)	2 (5.71)	0 (0)	1 (5.26)	1 (7.69)

Abbreviations: ICI, immune checkpoint inhibitor; AI, anti-angiogenesis inhibitor; NSCLC, non-small-cell lung cancer; TX, treatment; ECOG PS, Eastern Cooperative Oncology Group performance status; ADC, adenocarcinoma; SQC, squamous cell carcinoma.

**Table 2 cancers-16-00935-t002:** Baseline molecular characteristic of patients with docetaxel +/− ramucirumab treatment.

Genomic Mutations	Docetaxel Monotherapy (*n* = 178) (%)	Docetaxel + Ramucirumab (*n* = 203) (%)	Overall (*n* = 381) (%)
**Unknown**	15 (8.43)	13 (6.40)	28 (7.35)
**TP53 alterations**			
Positive TP53 alterations	69 (38.76)	98 (48.28)	167 (43.83)
Negative TP53 alterations	94 (52.81)	92 (45.32)	186 (48.82)
**KRAS alterations**			
Positive KRAS alterations	48 (26.97)	78 (38.42)	127 (33.33)
G12A	3 (1.69)	1 (0.49)	4 (1.05)
G12C	18 (10.11)	22 (10.84)	40 (10.50)
G12D	9 (5.06)	22 (10.84)	31 (8.14)
G12F	1 (0.56)	1 (0.49)	2 (0.52)
G12V	8 (4.49)	17 (8.37)	25 (6.56)
G13C	1 (0.56)	3 (1.48)	5 (1.31)
G13D	1 (0.56)	2 (0.99)	3 (0.79)
G12V+G13D	1 (0.56)	0 (0)	1 (0.26)
Q61L	0 (0)	2 (0.99)	2 (0.52)
Q61K	1 (0.56)	0 (0)	1 (0.26)
Q61H	2 (1.12)	1 (0.49)	3 (0.79)
Other KRAS alterations	3 (1.69)	7 (3.45)	10 (2.62)
Negative KRAS alterations	115 (64.61)	112 (55.17)	226 (59.32)
**EGFR alterations**			
Positive EGFR alterations	29 (16.29)	36 (17.73)	65 (17.06)
EGFR Del19/EGFR L858R(+/−T790M)	14 (7.87)	20 (9.85)	34 (8.92)
EGFR Exon20 ins	6 (3.37)	5 (2.46)	11 (2.89)
Other atypical EGFR mutations	2 (1.12)	0 (0)	2 (0.52)
Nonactionable EGFR alterations	7 (3.93)	11 (5.42)	18 (4.72)
Negative EGFR alterations	134 (75.28)	154 (75.86)	288 (75.59)
**STK11 alterations**			
Positive STK11 alterations	16 (8.99)	16 (7.88)	32 (8.40)
Negative STK11 alterations	147 (82.58)	174 (85.71)	321 (84.25)
**ERBB2 (HER2) alterations**			
Positive ERBB2 (HER2) alterations	11 (6.18)	15 (7.39)	26 (6.82)
Negative ERBB2 (HER2) alterations	152 (85.39)	175 (86.21)	327 (85.83)
**MET alterations**			
Positive MET alterations	16 (8.99)	8 (3.94)	24 (6.30)
MET amplification	8 (4.49)	7 (3.45)	15 (3.94)
MET exon14 skipping mutation	2 (1.12)	0 (0)	2 (0.52)
Other MET alterations	6 (3.37)	1 (0.49)	7 (1.84)
Negative MET alterations	147 (82.58)	182 (89.66)	329 (86.35)
**PIK3CA alterations**			
Positive PIK3CA alterations	8 (4.49)	11 (5.42)	19 (4.99)
Negative PIK3CA alterations	155 (87.08)	179 (88.18)	334 (87.66)
**BRAF alterations**			
Positive BRAF alterations	10 (5.62)	8 (3.94)	18 (4.72)
BRAF V600E	2 (1.12)	4 (1.97)	6 (1.57)
Other BRAF alterations	8 (4.49)	4 (1.97)	12 (3.15)
Negative KRAS alterations	153 (85.96)	182 (89.66)	335 (87.93)
**ALK alterations**			
Positive ALK alterations	7 (3.93)	4 (1.97)	11 (2.89)
ALK-EML4 fusion	1 (0.56)	3 (1.48)	4 (1.05)
Other ALK alterations	6 (3.37)	1 (0.49)	7 (1.84)
Negative PIK3CA alterations	156 (87.64)	186 (91.63)	342 (89.76)
**NTRK alterations**			
Positive NTRK alterations	3 (1.69)	4 (1.97)	7 (1.84)
NTRK1/2/3 gene fusion positive	0 (0)	1 (0.49)	1 (0.26)
Other NTRK alterations	3 (1.69)	3 (1.48)	6 (1.57)
Negative NTRK alterations	160 (89.89)	186 (91.63)	346 (90.81)
**ROS1 alterations**			
Positive ROS1 alterations	5 (2.81)	1 (0.49)	6 (1.57)
ROS1 translocation	2 (1.12)	1 (0.49)	3 (0.79)
Other ROS1 alterations	3 (1.69)	0 (0)	3 (0.79)
Negative ROS1 alterations	158 (88.76)	189 (93.10)	347 (91.08)
**RET alterations**			
Positive MET alterations	2 (1.12)	3 (1.48)	5 (1.31)
RET rearrangement	0 (0)	1 (0.49)	1 (0.26)
Other RET alterations	2 (1.12)	2 (0.99)	4 (1.05)
Negative MET alterations	161 (90.45)	187 (92.12)	348 (91.34)

Abbreviations: TP53, tumor protein 53; KRAS, Kirsten rat sarcoma virus; EGFR, epidermal growth factor receptor; STK11, serine/threonine kinase 11; ERBB2, receptor tyrosine–protein kinase erbB-2; MET, MET proto-oncogene; PIK3CA, phosphatidylinositol-4,5-bisphosphate 3-kinase catalytic subunit alpha; BRAF, V-Raf murine sarcoma viral oncogene homolog B; ALK, anaplastic lymphoma kinase; NTRK, neurotrophic tyrosine receptor kinase; ROS1, ROS proto-oncogene 1; RET, RET proto-oncogene.

## Data Availability

Data are contained within the article.

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
