# Peer review of "Clinical Benefit from Docetaxel +/− Ramucirumab Is Not Associated with Mutation Status in Metastatic Non-Small-Cell Lung Cancer Patients Who Progressed on Platinum Doublets and Immunotherapy"

_cancers, 2024, doi:10.3390/cancers16050935_

Round 1
Reviewer 1 Report
Comments and Suggestions for Authors
Thank you for submitting this interesting and informative manuscript to Cancers. I was pleased to receive it as a reviewer.
While your manuscript provides valuable insights into an important clinical topic, there are certain areas that could be refined to further enhance the quality and impact of the work. Here are some respectful suggestions that could potentially improve the paper if you choose to implement them:
Abstract
- Consider clarifying if comparisons between subgroups were pre-specified or exploratory.
Introduction
- Consider expanding the discussion of prior real-world data on efficacy of platinum/taxane doublets and genomic predictors after ICI failure to better contextualize the novelty of this study question.
- Consider summarizing the current standards of care for second line treatment of advanced NSCLC after immunotherapy failure. This will orient readers and highlight gaps in evidence that this study aims to address.
Methods
- Consider providing additional details on the platinum-based regimens used (specific doublet or triplet regimens, dosing, cycles, etc.)
- Consider inclusion of a CONSORT diagram depicting derivation of the final analytic cohorts
- Consider describing how sample size was determined. Was the study powered to detect differences between subgroups or just overall outcomes? This is relevant to interpret the results.
- Consider providing more details on the statistical analysis methods used for making comparisons between subgroups. Were they pre-specified or exploratory? Were adjustments made for multiple testing?
Discussion
- Consider further exploring potential reasons for observed efficacy differences between regimens (biological mechanisms, treatment duration etc.)
- Consider discussing generalizability of findings to broader advanced NSCLC populations outside the study sample. How might real-world practice differ?
- Consider commenting on whether findings may inform hypotheses for future randomized studies comparing platinum doublets to docetaxel +/- ramucirumab in this setting. This will highlight implications and next steps.
Overall, these suggested edits could further enhance transparency, methodologic rigor, and impact of this nice contribution to the literature. Your consideration of this input could ultimately strengthen the manuscript in support of publication.
Reviewer 2 Report
Comments and Suggestions for Authors
Nice work from Kang Qin et al. are describing an interesting article regarding the benefit of docetaxel with or without ramucirumab for patients regardless mutation status in metastatic non-small-cell lung cancer patients who progressed on platinum doublets and immunotherapy.
The aim of the study was to investigate whether cancer gene mutation status is associated with clinical benefits from docetaxel +/- ramucirumab than platinum/taxane-based.
There were 3 groups 1-received docetaxel monotherapy, 2- received docetaxel + ramucirumab, 3- platinum/taxane-based regimens.
The results showed that the platinum/taxane-based regimens were associated with a significantly longer mOS (13.00m,95%CI:11.20m-14.80m versus 8.40 m, 95% CI:7.12m-9.68m, 13 LogRank P=0.019) than docetaxel+/-ramcirumab. Also that none of the cancer gene mutations (TP53, KRAS, EGFR, STK11 and ERBB2) or PD-L1 expression was associated with PFS or OS.
Interesting research and information for us as oncologists.
Few comments:
Minor comments-
Section 1 is the introduction and not abstract.
Major comments-
Regarding 2nd line treating with platinum/taxane-based? What was the first line for patients with SQC? Thy were not treated with TAXOL+CARBO?+/- IO?
How could you give taxane-based for 2nd line of therapy if they progressed in that line?
Table 2- Docetaxel+/-Ramucirumab? This means that you don’t know if they received or didn’t receive Ramucirumab with Docetaxel? This line is misleading.
Could you please explain that.
Round 2
Reviewer 2 Report
Comments and Suggestions for Authors
I have nothing to add